# Screen of *Pinus massoniana* for Resistance to Pinewood Nematode: In Vitro Propagation and Evaluation of Regenerated Microshoots

**Jia-Yi Guo** [1,2,†], **Zi-Hui Zhu** [1,2,†], **You-Mei Chen** [1,2] and **Li-Hua Zhu** [1,2,*]

1    Collaborative Innovation Center of Sustainable Forestry in Southern China, Nanjing 210037, China; jyguo@njfu.edu.cn (J.-Y.G.); zhuzihui@njfu.edu.cn (Z.-H.Z.); chenyoumei@njfu.edu.cn (Y.-M.C.)
2    Institute of Forest Protection, College of Forest, Nanjing Forestry University, Nanjing 210037, China
*    Correspondence: lhzhu@njfu.com.cn
†    These authors contributed equally to this work.

**Abstract:** To accelerate breeding of *Pinus massoniana* Lamb. resistance to pine wilt disease (PWD), caused by the pinewood nematode (PWN), *Bursaphelenchus xylophilus*, a protocol was established for the in vitro propagation of *P. massoniana* and the evaluation of resistance of regenerated microshoots from different clones to PWN. Axillary bud induction was achieved by culturing cotyledonary node explants from 3-week-old seedlings in Gupta and Durzan (DCR) medium that was supplemented with 4 mg L$^{-1}$ 6-benzyladenine (BA) and 0.2 mg L$^{-1}$ α-naphthaleneacetic acid (NAA). Explants with induced buds were transferred to DCR medium without a plant growth regulator to facilitate elongation. Stem segments from elongated shoots were used as propagules for further shoot multiplication. Six-month-old regenerated shoots that met the requirements for a nematode resistance test were inoculated with aseptic PWN (500 PWNs/shoot). The wilting rate varied between clones from 20% to 100%, 18 days after inoculation. Except for Clone 227, which showed the highest resistance with a wilting rate of 0%, other clones showed wilting to various degrees 30 d after inoculation. The number of nematodes that were recovered from Clone 227 was significantly lower than from other clones. This study promotes the resistance breeding of *P. massoniana* to pine wilt disease and provides an effective method to study the host/pathogen interaction between PWN and *P. massoniana*.

**Keywords:** *Pinus massoniana*; axillary budding; genetic stability; *Bursaphelenchus xylophilus*; nematode resistance

## 1. Introduction

Masson pine (*Pinus massoniana* Lamb.) is an evergreen coniferous species native to central and southern China [1]. It is regarded as an important species for afforestation purposes and has been highly exploited for its timber and natural resin [2]. However, *P. massoniana* is highly susceptible to pine wilt disease (PWD), which is caused by the pinewood nematode (PWN), *Bursaphelenchus xylophilus*. The first occurrence of PWD was reported in 1905 in Japan; PWN was only identified as the causal agent in 1971 [3]. Until recently, most of the researchers thought that PWN was the only known causal agent causing pine tree wilting [4–7], but some others think that this disease is induced by both PWN and the pathogenic bacteria that it carries [8]. Some experiments indicated a mutually beneficial, symbiotic relationship between PWN and its associated bacteria [9]. In contrast, some reports indicated that bacteria might be endophytes which attach to the nematode cuticle, not taking part in PWD [7,10]. Although various control methods have been used to eradicate PWD, this disease has continued to expand over recent decades and has caused the destruction of hundreds of millions of pine trees, particularly in East Asia and southern Europe, which has resulted in enormous economic losses and profound ecological

consequences [11–17]. In 1982, the disease was detected in pine forests in Nanjing, Jiangsu Province [18], and then has spread to 19 provinces and municipalities in 40 years [19].

The rapid spread of PWD has severely affected the economy and social sustainable development in China. For instance, the death of a large number of *P. massoniana* caused by PWD in the Three Gorges reservoir region has devastated local ecosystem services [20]. Once an infection is established, most countermeasures for controlling the disease depend on chemical methods which are not economically effective. In comparison, breeding pine species for resistance to PWD is considered to be a fundamental strategy for prevention [12]. In southwestern Japan, breeding programs for pine species were initiated in 1978, mainly focusing on *Pinus densiflora* and *Pinus thunbergii*, and resistant plantations were successfully established in 1987 [21]. In recent years, breeding of PWD-resistant *P. massoniana* has been carried out in China [22]. However, plantations initiated from seed stock are not productive, which limits mass production of this species in the short term [23].

In vitro propagation is the main regeneration pathway of plants, and it can be an appropriate mean for clonal production of superior tree genotypes. A large number of regenerated plantlets can be obtained using small pieces of plant tissue or organs as the starter material [24]. Propagation based on organogenesis and somatic embryogenesis has been demonstrated on 40 species in the genus *Pinus* [25–28]. Meanwhile, in vitro reproduction of *P. massoniana* with similar micropropagation protocols has been well documented. In vitro induction of adventitious buds from mature embryos was reported by Wu et al. for the first time [29]. Three regenerated plantlets were obtained through somatic embryogenesis from mature zygotic embryos of *P. massoniana* by Huang et al. [30]. Since then, numerous studies have resulted in efficient in vitro propagation protocols for *P. massoniana*, using zygotic embryos, seedling explants, and explants from mature trees [31–35].

True-to-type clonal fidelity is fundamental to the successful in vitro propagation of tree species [36]. However, several levels of variation among regenerated plants have been reported, comprising morphological, biochemical, and genetic variations [37,38]. Therefore, it is necessary to monitor clonal variation during in vitro propagation in order to guarantee the genetic stability of cloned plants. The informative DNA markers have been widely used to evaluate the genetic variation at the molecular level. Long-term (more than 10 years) micropropagated shoots of *P. thunbergii* were determined to be genetically stable through random amplified polymorphic DNA (RAPD) markers, in which short, medium, and long morphotypes of shoots were randomly collected for DNA fingerprints [36]. RAPD was also successfully carried out for *Pinus elliottii* to analyze somaclonal variation [39]. During the development of molecular marker technology, simple sequence repeats (SSRs) which have reproducibility, abundant polymorphisms, and codominant inheritance predominated [40]. SSRs were reported to monitor mutation events during somatic embryogenesis in *Pinus pinaster*, *P. elliottii*, and *P. massoniana*; no correlation was evident between genetic stability and abnormal phenotype [41–43]. To our knowledge, there have been no reports on variation in micropropagated *P. massoniana* via axillary budding and adventitious buds determined by SSRs until now.

As a valuable biotechnological tool, in vitro propagation assists the breeding of superior tree genotypes, and it has attracted the attention of plant pathologists in the search for effective ways to prevent and control tree diseases. Many researchers have studied the response of conifer tissues to specific diseases by directly exposing tissue cultures to live pathogenic organisms or their extracts, and this approach may speed up resistance breeding [44]. For example, Terho et al. inoculated embryogenic cell lines of *Pinus sylvestris* with live fungal spores from scleroderris cankers caused by *Gremmeniella abietina* [45]. Although the resistance of the culture could not be demonstrated, a difference in growth rates and glucosamine levels formed in response to fungal infection were detected between the cell lines. Nagy et al. treated embryogenic cell lines of *Picea abies* with spores of the blue-stain fungal pathogen *Ceratocystis polonica* and the butt rot pathogen *Heterobasidion annosum* [46]. The response of cell cultures was similar to trees which differ in susceptibility to these diseases. *Picea abies* cell cultures were inoculated with spores of *C. polonica* by

Phillips et al., and the defense reactions and gene expression patterns were tested [47]. In another study, an axenic host/pathogen system was developed to study the role played by PWN and their associated bacteria in PWD development by inoculating aseptic PWN or bacteria to micropropagated shoots of *P. densiflora* under in vitro conditions [48]. This system was then adopted to evaluate the resistance of regenerated microshoots of *P. densiflora* to PWN [49]. A similar maritime pine/nematode co-culture system was established by Faria et al., and the effects of phytoparasites on shoot structure, water content, and volatiles were evaluated in vitro [6]. Symptoms similar to those under natural infection conditions were observed.

Verification of resistance status in nursery and field trials is necessary when presumptive resistant plants are produced by in vitro propagation. Unfortunately, this vital step is frequently neglected [44]. However, this does not mean that in vitro propagation is of little value for investigating tree diseases as it can facilitate the rapid propagation of selected plants for the study of tree diseases at any time of year, which can increase the efficiency of breeding resistance.

In vitro co-culture of host and parasite can be a useful system to study plant/nematode interactions because it eliminates variables that arise from environmental conditions, it excludes associated microbiota, and it can facilitate direct observation of plant/nematode responses in a controlled-contaminant-free environment, which is very difficult to achieve in field conditions [6,48]. At present, there are no evaluations of PWN resistance in *P. massoniana* plantlets derived from in vitro propagation. The aim of our study was to establish a simple protocol for in vitro propagation of *P. massoniana* via axillary budding and adventitious buds where the genetic stability was analyzed and evaluate the PWN resistance of multiple clones by in vitro inoculation of micropropagated shoots with aseptic PWN. Our work also provided a suitable system for PWD phytopathological research.

## 2. Materials and Methods

### 2.1. Plant Materials and PWN

Mature seeds from open-pollinated *Pinus massoniana* trees were collected from Guangdong Province in China in 2019. All seeds were kept at 4 °C until used.

The PWN isolate AMA3cl is a strong virulent and full-sibling mating inbred line, which was used to inoculate regenerated shoots in vitro [48]. Bacteria-free nematodes were subcultured on *Botrytis cinerea*. Prior to inoculation, the aseptic nematodes were stored in sterile water at 4 °C.

### 2.2. Seed Disinfection and In Vitro Germination

The seed disinfection method was a modification of that described by Zhu et al. [33]. Seeds of *P. massoniana* were washed with sterile water for 1 h and then placed in a centrifuge tube with small holes around the base. Then, the seeds were surface sterilized by soaking in 75% alcohol for 30 s followed by immersion in 30% $H_2O_2$ solution for 20 min. During this period, the seeds were stirred with sterilized tweezers to ensure full disinfection. After rinsing with distilled water, seeds were dried with sterilized filter paper and then cultured on water agar medium. Each Petri dish containing 10 seeds was maintained in darkness at 25 °C. One week later, the aseptic germinated seeds were transferred to Gupta and Durzan (DCR) (1985) medium supplemented with 30 g $L^{-1}$ sucrose and 1.0 g $L^{-1}$ activated carbon (AC) (pH = 5.8), cultured under light at 25 °C [50]. Unless otherwise stated, the experiments were carried out at 25 ± 2 °C under a 16 h photoperiod with a light intensity of 36 lmol $m^{-2}$ $s^{-1}$ from cool white fluorescent illumination.

### 2.3. Axillary Bud Formation and Elongation

The protocol for *P. densiflora* published by Zhu et al. was used [49]. Because cotyledonary node explant contains meristematic cells, they were excised from seedlings and cultured on DCR medium containing 4 mg $L^{-1}$ 6-benzyladenine (BA) and 0.2 mg $L^{-1}$ a-naphthaleneacetic acid (NAA) under light for axillary bud induction. After culturing for

4 weeks, explants with induced buds were transferred to hormone-free DCR medium but supplemented with 20 g L$^{-1}$ sucrose, 0.5 g L$^{-1}$ inositol, and 0.75 g L$^{-1}$ activated carbon to promote shoot development and elongation. The pH of the medium was adjusted to 5.8 before adding carrageenan and autoclaving at 121 °C for 20 min. Four weeks later, strong buds were selected and transferred to fresh DCR medium for further shoot elongation.

### 2.4. Shoot Multiplication

Elongated shoots were cut into stem segments of 5–6 mm, and then cultured on DCR medium containing 2 mg L$^{-1}$ BA, 0.2 mg L$^{-1}$ NAA, 30 g L$^{-1}$ maltose, 0.5 g L$^{-1}$ inositol, and 0.45 g L$^{-1}$ hydrolyzed casein to promote bud proliferation. After 4 weeks culture on this bud multiplication medium, explants with induced buds were transferred to shoot elongation medium as described previously. This procedure was repeated routinely at about 10-week intervals. All shoots generated from the same seed were described as a clone.

### 2.5. DNA Isolation and Quantification

For DNA extraction, 12-cycle-regeneration microshoots from five clones (Clone 202, Clone 207, Clone 222, Clone 226, and Clone 253) and six microshoots were randomly chosen from each clone. Microshoots were ground in liquid nitrogen. DNA was extracted from each sample using a Bioteke DP3111 Plant Genomic DNA Extraction Kit. DNA concentration and purity were measured using ultraviolet spectrophotometrically and agarose gel electrophoresis.

### 2.6. SSR Amplification and Fragment Analysis

Six primer pairs, from available nuclear microsatellite primers already tested in *P. massoniana*, were chosen for this experiment [43,51]. These primers were selected based on good repeatability and high quality. The pine nuclear microsatellite loci analyzed were P.Ma 43, P.Ma 51, P.Ma 65, P.Ma 77, P.Ma 95, and P.Ma 96. The information of the primers used is shown in Table 1. Each forward primer was labeled with Tsingke's Gold Mix (green) (Cat. No. TSE101). Electrophoresis and detection of bands were carried out with ABI3730 sequencer. GeneMapper 5 software was used to analyze the peak patterns. The allele was considered to be mutated when a mismatch of more than 2 bp was observed comparing with the size of the original allele.

**Table 1.** PCR analysis of SSR loci in *P. massoniana* showing loci identification, forward (F), and reverse (R) primers used for amplification of respective SSRs, length, annealing temperature, and number of cycles in 30 samples tested.

| Locus | Primers (5'-3') | Length (bp) | Annealing Temperature (°C) | Number of Cycles | Identification |
|-------|-----------------|-------------|---------------------------|------------------|----------------|
| P.Ma43 | F:GCAACCTCCATATTTCACTT R: CTTTCCAATCTTCCCTTACA | 234 | 51 | 26 | KC146075 |
| P.Ma51 | F: ACGCACGGATAAGATTGTG R: ATCAAGTTACCCTCATTTGGA | 227 | 55 | 24 | KC146077 |
| P.Ma65 | F: AAGGCACTCGATCTCCTC R: TGACCTGCTTCTACACCC | 248 | 60 | 24 | KC146078 |
| P.Ma77 | F: GACCGTACAACACTCACTTGA R: CCTCTTTCCCTTGTCCTG | 323 | 51 | 24 | KC146081 |
| P.Ma95 | F: CTACCGATGCGATAAGGG R: ACTCGTGACTGCGACAATAC | 303 | 52 | 24 | KC146084 |
| P.Ma96 | F: TGACCCAATAGACTCCCTC R:AGACCTATCTAAGCACAACCC | 260 | 52 | 25 | KC146085 |

### 2.7. Acquisition of Sterilized Nematodes

Sterilized nematodes were obtained as described by Zhu et al. [48]. In asepsis, the nematode suspension was placed on the cover slip in a Petri dish, and the nematodes were thrown away after laying eggs. The eggs on the surface of the cover slip were washed

several times with sterile water and immersed in 15% $H_2O_2$ for 60 min at 25 °C for surface sterilization. Nematode eggs were washed with sterile water 3 times and kept in the dark on the mycelia of *B. cinerea* on the PDA medium.

The propagated nematodes were obtained with a Baermann funnel in asepsis and nematode sterilization was tested in nutrient broth (NB) liquid medium for more than 7 days. Bacterium-free PWNs were used to inoculate regenerated microshoots in vitro.

### 2.8. In Vitro Tolerance Assay with Nematodes

The in vitro inoculation method was described previously by Zhu et al. [49]. To inoculate the regenerated shoots with sterile nematodes, the apical buds of regenerated microshoots were excised with a cotton ball placed at the wound. The interface between the cotton ball and the wound was inoculated with 50 µL of aseptic nematode suspension containing 500 nematodes. Shoots inoculated with sterile water were used as a control. Each clone had 10 replicates. The symptom changes of shoots (chlorosis and shrinkage) were observed every 4 days during a 30-day period, and withered plantlets were recorded. More than half of the shoots with yellow needles were defined as wilting. The wilting rate of each clone was the ratio of the number of wilted shoots to the total number of shoots.

Thirty days after inoculation, nematodes were recovered by a Baermann funnel from the inoculated shoots and the culture medium. For isolation of nematodes, the inoculated shoots were cut into 2–4 mm segments. Nematodes were collected in centrifuge tubes after standing for 12 h. The number of nematodes recovered from the shoots and culture medium were counted under the microscope.

### 2.9. Statistical Analysis

GraphPad prism 6.01 software (GraphPad Software, Inc., La Jolla, CA, USA) was used for variance analysis. The data were expressed as the mean ± standard deviation. Statistical analyses used were one-way ANOVA (with different clones as factors), Duncan's test (significant statistical difference among clones), and the LSD-*t* test (significant statistical difference among clones). Test data were homoscedasticity and normality.

## 3. Results

### 3.1. Formation of Axillary Buds and Elongation of Shoots

For seed disinfection, the protocol adopted in this study provided low contamination rates (2%) and high germination rates (68%).

Intercotyledonary axillary bud initiation and development were observed in 90% of explants 4 weeks after culturing on DCR medium supplemented with 4 mg $L^{-1}$ BA and 0.2 mg $L^{-1}$ NAA. The average number of axillary buds from each explant was 3.5, with a maximum of 8 buds.

Shoots elongated significantly after being transferred into hormone-free DCR medium containing 0.75 g $L^{-1}$ activated charcoal. Following 45 days culture, significant differences in shoot lengths derived from different clones were observed (Figure 1). The shoot lengths of Clone 207 (4.8 ± 0.3) were similar to that of Clone 226 (4.7 ± 0.2), followed by Clone 8 (4.6 ± 0.2) and Clone 253 (4.6 ± 0.3), which were both higher than other clones (Figure 2).

### 3.2. Shoot Proliferation

Elongated shoots were cut into 5–6 mm segments and transferred to DCR medium supplemented with 2 mg $L^{-1}$ BA and 0.2 mg $L^{-1}$ NAA for further multiplication. Four weeks after culturing, axillary meristems sprouted along shoots and axillary bud proliferation varied at the clone level (Figure 3). The explants generating buds were considered to be responsive. The highest average responsive frequency appeared in Clone 226 with 77.8% explants followed by Clone 207 with 72.1% and Clone 8 with 66.7% (Table 2). The average number of buds per explant of Clone 207 was the highest (5.3 ± 0.6), followed by Clone 253 (4.3 ± 0.6) and Clone 226 (4.0 ± 1.0). Once these axillary buds were transferred to DCR medium with 0.75 g $L^{-1}$ activated charcoal, they showed faster growth than at

the axillary bud induction stage. Six weeks later, they were used as explants for further proliferation. During each periodic subculture, the new shoots excised from the main mass were transferred to fresh medium to obtain more biomass. After multiple subcultures (12 months), Clone 207 yielded the most with 826 shoots, followed by Clone 253, Clone 226, and Clone 8, with 328, 294, and 265 shoots, respectively.

**Table 2.** Axillary buds induced from microshoots of nine clones.

| Clone Code | Responsive Explants (%) | Average No. of Buds/Explant | Total Buds after 12 Months of Proliferation |
|---|---|---|---|
| 8 | 66.7 | 3.7 ± 1.5 abc | 265 |
| 115 | 35.7 | 2.0 ± 1.0 cde | 98 |
| 202 | 57.1 | 3.3 ± 1.5 bcd | 162 |
| 207 | 72.1 | 5.3 ± 0.6 a | 826 |
| 220 | 30.0 | 1.7 ± 0.6 de | 72 |
| 222 | 41.7 | 3.7 ± 0.6 abc | 173 |
| 226 | 77.8 | 4.0 ± 1.0 ab | 294 |
| 227 | 38.5 | 1.0 ± 0.0 e | 45 |
| 253 | 60.8 | 4.3 ± 0.6 ab | 328 |

Data represent mean ± SD. Different letters indicate a significant difference ($p < 0.05$ by Duncan's test).

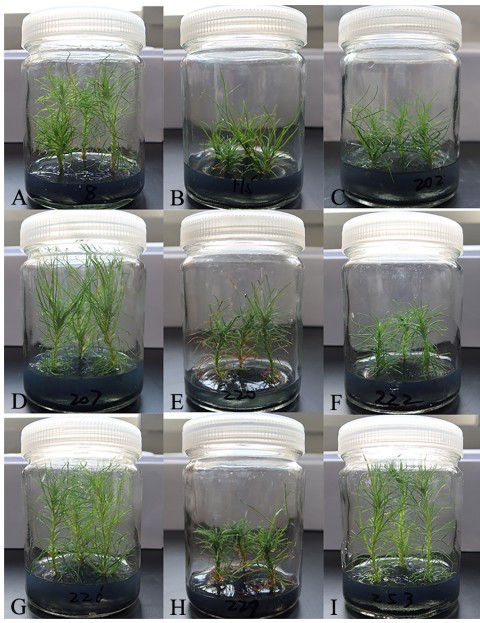

**Figure 1.** Elongated shoots cultured on Gupta and Durzan (DCR) medium with 0.75 g l$^{-1}$ activated charcoal. (**A**): Clone 8, (**B**): Clone 115, (**C**): Clone 202, (**D**): Clone 207, (**E**): Clone 220, (**F**): Clone 222, (**G**): Clone 226, (**H**): Clone 227, (**I**): Clone 253.

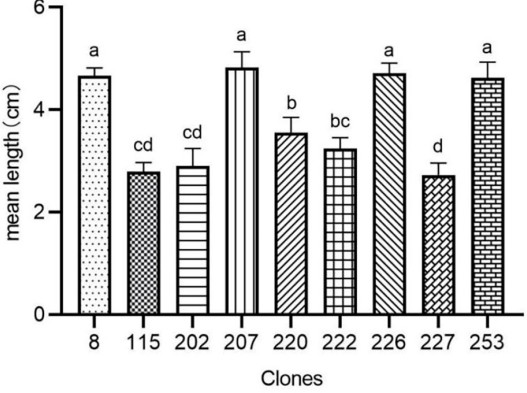

**Figure 2.** Microshoot length of nine selected clones. Different letters indicate significant differences in shoot length among different clones at $p < 0.05$.

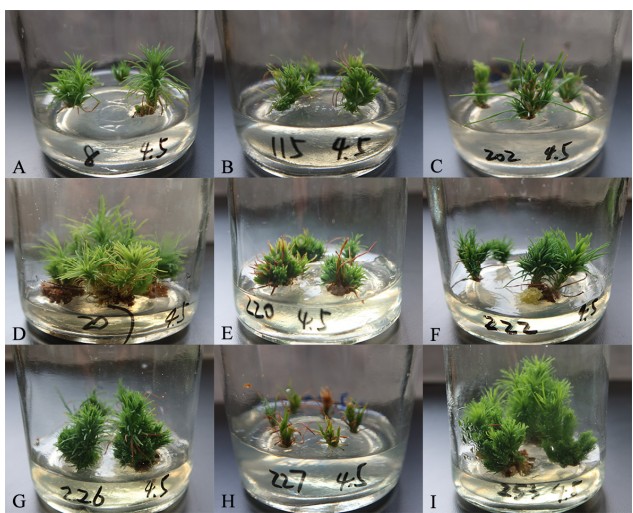

**Figure 3.** In vitro multiplication of *P. massoniana* from stem segments by axillary bud proliferation. Stem segments cultured on Gupta and Durzan (DCR) medium with 2 mg L$^{-1}$ BA and 0.2 mg L$^{-1}$ NAA after 3 weeks. (**A**): Clone 8, (**B**): Clone 115, (**C**): Clone 202, (**D**): Clone 207, (**E**): Clone 220, (**F**): Clone 222, (**G**): Clone 226, (**H**): Clone 227, (**I**): Clone 253.

### 3.3. Genetic Stability in Regenerated Shoots

A total of thirty regenerated microshoots from five clones were analyzed at six SSR loci. Except for six microshoots from Clone 226, which showed no variation in amplification profiles, seven (29.2%) of the twenty-four microshoots from four clones showed variation at tested loci. The size variation at five loci (P.Ma43, P.Ma51, P.Ma65, P.Ma95, and P.Ma96), detected in four clones (202, 207, 222, and 253), was showed in Table 3. The average variation rate of per locus was 6.7%. The highest mutation percentage (16.67%) was observed in the P.Ma51 locus (Table 4), where genetic variation was found in five microshoots, from Clone 202, 222, and 253. No mutation was observed at P.Ma77.

**Table 3.** Fragment lengths of alleles in plantlets of *P. massoniana*.

| Clone | Individuals with Mutated Alleles | Loci | Original Alleles | Mutated Alleles |
|---|---|---|---|---|
| | A | P.Ma43 | 232/241 | 232/232 |
| | | P.Ma51 | 215/224 | 215/239 |
| | | P.Ma96 | 264/264 | 240/260 |
| 202 | B | P.Ma51 | 215/224 | 215/239 |
| | C | P.Ma51 | 215/224 | 224/239 |
| | | P.Ma65 | 244/244 | 244/251 |
| | D | P.Ma43 | 232/241 | 232/232 |
| 207 | A | P.Ma96 | 273/273 | 271/271 |
| | | P.Ma51 | 215/224 | 224/224 |
| 222 | A | P.Ma65 | 240/244 | 244/244 |
| | | P.Ma95 | 288/288 | 288/292 |
| 253 | A | P.Ma51 | 215/224 | 215/239 |

**Table 4.** Variation frequency (%) of mutated shoots of *P. massoniana* at P.Ma43, P.Ma51, P.Ma65, P.Ma77, P.Ma95, and P.Ma96 loci.

| | Loci | | | | | |
|---|---|---|---|---|---|---|
| | **P.Ma43** | **P.Ma51** | **P.Ma65** | **P.Ma77** | **P.Ma95** | **P.Ma96** |
| No. analyzed shoots | 30 | 30 | 30 | 30 | 30 | 30 |
| No. mutated shoots | 2 | 5 | 2 | 0 | 1 | 2 |
| Variation frequency (%) | 6.7 | 16.7 | 6.7 | 0 | 3.3 | 6.7 |

### 3.4. Acquisition of Sterilized Nematodes

The fungal mats placed by nematode eggs disappeared from the medium 2–3 weeks later. The absence of bacteria was checked in NB liquid medium for more than 7 days after surface sterilization, and all populations were bacterium-free.

### 3.5. PWN Tolerance of Regenerated Shoots

Regenerated shoots inoculated with aseptic PWNs showed similar symptoms to those observed in the field. Twenty days after inoculation, all clones, except Clone 227, showed obvious wilting symptoms caused by infection of PWNs; i.e., the needles were discolored, yellowing, and browning, and whole explants withered and died gradually (Figure 4). However, control shoots inoculated with sterile water remained green and in good condition (Figure 4). For Clone 227, sprouting axillary meristems were observed in some inoculated shoots (Figure 4).

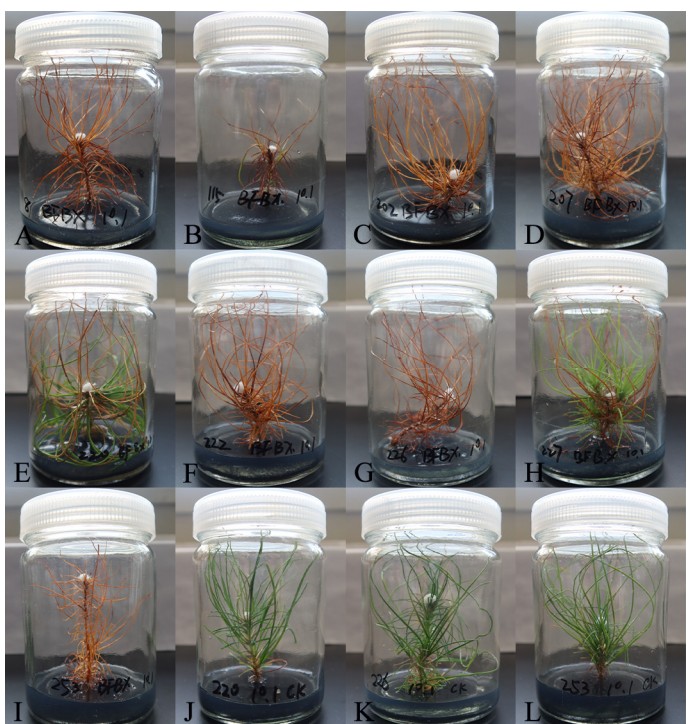

**Figure 4.** Wilting symptoms of different clones of *P. massoniana* 30 days after inoculation with sterile pinewood nematode. (**A**): Clone 8, (**B**): Clone 115, (**C**): Clone 202, (**D**): Clone 207, (**E**): Clone 220, (**F**): Clone 222, (**G**): Clone 226, (**H**): Clone 227, (**I**): Clone 253, (**J–L**) Clone 220, Clone 226, and Clone 253 inoculated with sterile water.

The wilting rate varied substantially between different clones (Figure 5). Twenty-eight days after inoculation, Clone 227 showed the highest resistance to PWN with a wilting rate of 0%. Only a few pine needles showed chlorosis, but not to the point of wilting (i.e., more than 50% of the needles yellowing), followed by Clone 220 which had a wilting rate of 30%. Clone 8, Clone 202, and Clone 207 had a wilting rate of 60%. The wilting rates of clones 115 and 222 were about 70% after 20 days from inoculation. Clones 226 and 253 showed obvious susceptibility to PWNs, and, after 28 days from inoculation, the wilting rates were as high as 80% and 90%, respectively (Figure 5).

Nematodes were recovered from shoots of each clone 30 days after inoculation. No nematode was recovered from control shoots. The number of nematodes recovered from clone 253 was the highest (2413; range: 830–3913; $n = 3$), which was higher than that of all other clones ($p < 0.05$) (Figure 6A). Clone 8 had the next highest number of recovered nematodes, with an average of 2110 PWNs per shoot (range: 925–3206; $n = 3$). The lowest number of nematode recoveries was recorded in Clone 227 (194; range: 142–243; $n = 3$).

Nematodes were recovered from the corresponding control medium. Nematodes that were recovered from the culture medium of Clone 253 had the highest number (7446; range: 6579–8190, *n* = 3), followed by Clone 220 (4032; range: 3366–5309, *n* = 3) (Figure 6B). The number of nematodes recovered from the culture medium of Clone 227 was the lowest (645; range: 429–1026, *n* = 3) (Figure 6B).

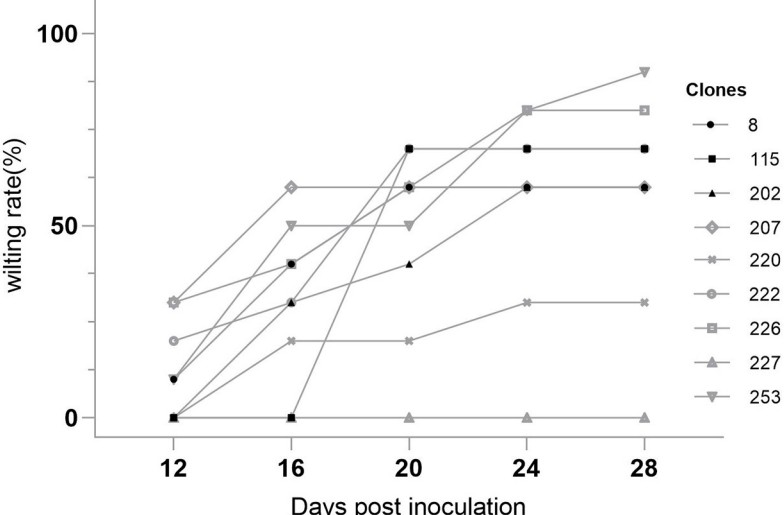

**Figure 5.** Wilting rates of nine clones of *P. massoniana* after inoculation with sterile pinewood nematode.

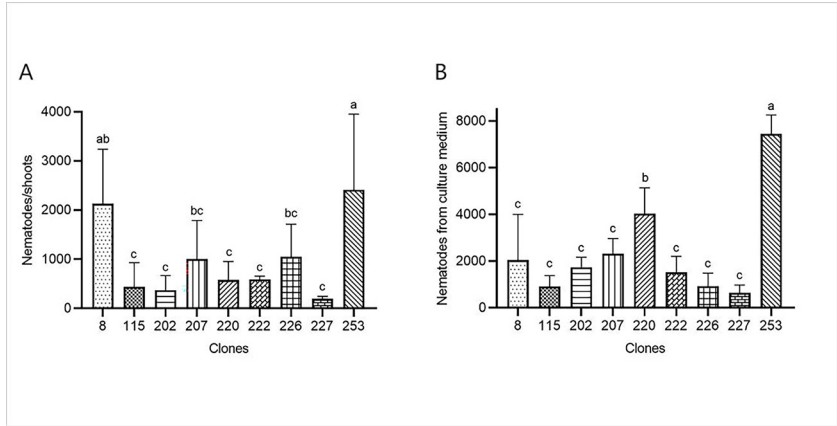

**Figure 6.** Number of nematodes recovered from microshoots of different clones after inoculation with aseptic pinewood nematode (**A**) and from corresponding culture medium of different clones (**B**). Bars with different letters are significantly different at *p* < 0.05 according to Duncan's multiple range tests.

## 4. Discussion

In vitro propagation by direct organogenesis is the optimum method for rapid multiplication, and it can contribute to the generation of true-to-type plants, which has been applied to many pine species, such as *Pinus strobus*, *Pinus taeda*, and *P. pinaster* [52–55]. Although most research in in vitro propagation of *P. massoniana* has been conducted using somatic embryogenesis, the propagation pathway via direct organogenesis has also been reported [29,33,56–58]. In this study, an in vitro propagation system for *P. massoniana* was successfully established where a cotyledon-hypocotyl was cultured as the initial explant and multiplication was achieved by inducing axillary buds.

BA, as a cytokinin, is the critical factor affecting germination of axillary buds during the regeneration of *P. massoniana*. As reported, the induction of buds from meristems was retarded when *P. massoniana* explants were cultured on the basal medium, while axillary buds were observed after supplementation of the medium with BA [33]. In addition,

the presence of NAA was beneficial in the propagation procedure. It has been reported that adding BA and NAA in an appropriate proportion to a culture medium can more effectively promote the development of axillary buds [49]. The highest induction frequency of adventitious buds reached 99.3% when Zhang et al. cultured adventitious buds on DCR medium supplemented with 0.5 mg L$^{-1}$ BA and 0.05 mg L$^{-1}$ NAA [31]. An average of 3.9 buds were induced from 92% of explants after culture on Gresshoff and Doy (GD) medium including 2 mg L$^{-1}$ BA and 0.2 mg L$^{-1}$ NAA [33]. Wang et al. reported plantlet regeneration in vitro from mature trees of *P. massoniana*, and the mean number of buds per explant was 4.8 with an average length of 7.1 cm after 120 days culture [24]. In our study, cotyledonary node explants were cultured on a medium supplemented with 4 mg L$^{-1}$ BA and 0.2 mg L$^{-1}$ NAA for axillary bud induction; then, shoots after elongation were excised and transferred to a medium containing 2 mg L$^{-1}$ BA and 0.2 mg L$^{-1}$ NAA for proliferation. In this treatment, an average of 5.3 buds were induced from 72.1% explants of Clone 207.

The supplementation of activated carbon (AC) in the elongation medium was considered crucial for *P. massoniana* shoot growth [24]. As reported, AC could absorb inhibitory compounds harmful to shoot growth in culture medium, which may be the reason why AC could promote bud elongation [49,59]. Although the impact of AC on plant growth depends on the species and materials used, the promotive effect of AC on shoot elongation of *P. massoniana* has also been confirmed by Yao et al. [57]. In our study, the shoots elongated rapidly on the culture medium supplemented with AC; Clone 207 had an average length of 4.8 cm after 45 days culturing.

The evaluation of genetic stability during in vitro propagation is necessary for scale production of pine species. In the process of tissue culture, genetic instability of gymnosperms is easily induced by various factors, including a long culture period and high growth regulators [60,61]. Simple sequence repeats (SSRs) have been powerful tools to investigate genetic variability in the DNA sequences of plants. Marum et al. analyzed embryogenic cell lines of *P. pinaster*, and genetic variation was detected at seven SSR loci [41]. In their study, variation was seen in 5 of the 52 emblings in amplification profiles for tested loci, and the average variation rate per locus was 2.7%. In *P. elliottii*, 35% (5/14) of the regenerated emblings carried the mutated alleles, and the average rate of variation per locus was 7.1% [42]. Brug et al. reported that mutant maternal alleles were detected in ~40% of the embryonic cell lines of *P. sylvestris* [62]. Hazubska-Przybył et al. found changes of the DNA in plant material from 80% (8/10) of tested embryogenic lines of *P. abies* and from 52.6% (10/19) of embryogenic lines of *P. omorika* [63]. All callus cultures obtained from Siberian larch (*Larix sibirica* Ledeb.) megagametophytes contained new mutations in one or more microsatellite loci [64]. Xia et al. reported that 25% (4/16) of emblings showed variation, and the average variation rate per locus was 2.78% [43]. In contrast to the somatic embryogenesis system, the mutation risk in organogenesis system may be relatively low. Goto et al. reported that no somaclonal variation was detected within the microshoots of *P. thunbergii* which were cultured for more than 10 years [36]. Tang et al. also reported that no aberration RAPD banding patterns were detected among the in vitro propagated plantlets of *P. taeda* [65]. In our study, the variation of regenerated microshoots of *P. massoniana* at six SSR loci was monitored, where 23.3% shoots showed variation and the mean variation percentage per locus was 6.7%.

Tissue culture can produce a large number of regenerated plants, and it also provides a method for tree disease research, which makes an important contribution to disease resistance breeding [44]. To date, this method has been extensively used to evaluate the disease resistance in *Pinus* species. Cheng and Ye reported significant differences in several clones of *P. elliottii* with regards their resistance to brownspot (*Lecanosticta acicola*) [66]. In another study, the initial selection of superior genotypes resistant to PWD was achieved by the establishment of the Japanese red pine/nematode co-culture system [67]. Callus for *Pinus* species was used to evaluate the disease resistance in vitro; for instance, Ragazzi et al. reported differences in some *Pinus* species to resistance against blister rust (*Cronartium flaccidum*) which were observed by variations in colony growth on callus. The colonies grew

faster on callus from sensitive pine species than from resistant species [68]. In our study, the resistance to PWD was preliminarily evaluated by inoculating PWN on regenerated microshoots of *P. massoniana* under aseptic conditions and then by comparing the wilting rate of regenerated plantlets from different clones and measuring the recovery rate of nematode populations recovered from the inoculated plantlets. The wilting rate of Clone 227 and the number of nematodes recovered from Clone 227 were significantly lower than that of other clones, which indicated that this clone had a relatively high resistance to PWD. Similar observations have previously been reported by Zhu et al. [49]. In their study, Clone 8-4 exhibited high resistance to PWD by the lower wilting rate and lower nematode number compared to other clones. Resistant plants can be effectively obtained by in vitro screening. Faria et al. reported that maritime pine/PWN co-culture is an adequate biotechnological tool to study PWD, capable of evaluating the effect of nematotoxics addition in a host/parasite culture system [6]. Observation and recording of the influence of a pathogen on a host can greatly accelerate the process of disease study in conifers.

Although artificial inoculation has been applied to create conditions for disease occurrence and the resistance of plants can be assessed in the early stages of infection, field tests remain the benchmark for determining the resistance of plants [44]. Qu et al. tested and ranked the resistance of different potato varieties to blight caused by *Fusarium oxysporum* [69]. The results of field and laboratory tests were generally consistent. Luo et al. conducted the resistance test by treating tomato regenerated seedlings at different stages with antibiotics and herbicide, but this method was limited to the original screening, and field tests were critical [70]. In our study, two clones (Clones 207 and 220) showed relatively high resistance to PWD, but the results need further field assays to verify.

The pathogenic agent causing pine wilt disease is still controversial. It was originally thought that this disease was only attributed to PWN. Following the inoculation with aseptic PWN, *P. massoniana* and *P. densiflora* wilted which meant the aseptic nematodes did not lose their pathogenicity [48,49]. Nevertheless, the symbiotic bacteria carried by nematodes may contribute to PWD [10]. As reported, inoculation with sterile PWN and bacteria isolated from nematodes did not lead to pine branch wilting, but pine branches showed wilting after inoculation with a mixture of PWN and symbiotic bacteria [71]. In our experiment, aseptic PWN inoculation led to the wilting of several clonal microshoots raised from the tissue culture system, which was consistent with the results of Faria et al. [6] and Zhu et al. [48,49].

## 5. Conclusions

In conclusion, we have developed an effective protocol for the rapid propagation of *P. massoniana* from the initiation of axillary buds via direct organogenesis. The resistance of regenerated shoots to PWN was evaluated in vitro. This method can facilitate breeding of *P. massoniana* resistance to PWD. For the most PWN tolerant clone 227 obtained in this study, periodic subcultures will be continued to increase its biomass. Rooting and acclimatization of regenerated plantlets will be carried out in the next step, hoping to translate regenerated plantlets from in vitro to ex vitro. On this basis, plantlets were subjected to field tests to verify their resistance. Once the clone with resistance to PWN is obtained, it becomes possible to put this clone into mass production. Moreover, the in vitro co-culture system established in this study is beneficial for the study of the pathogenic mechanisms of pine wilt disease.

**Author Contributions:** Conceptualization, L.-H.Z.; methodology, J.-Y.G., Z.-H.Z. and Y.-M.C.; validation, J.-Y.G., Z.-H.Z. and Y.-M.C.; formal analysis, J.-Y.G., Z.-H.Z. and Y.-M.C.; investigation, J.-Y.G., Z.-H.Z. and Y.-M.C.; resources, L.-H.Z.; data curation, J.-Y.G., Z.-H.Z. and Y.-M.C.; writing—original draft preparation, J.-Y.G., Z.-H.Z. and Y.-M.C.; writing—review and editing, J.-Y.G. and L.-H.Z.; visualization, J.-Y.G., Z.-H.Z. and Y.-M.C.; supervision, L.-H.Z.; project administration, L.-H.Z.; funding acquisition, L.-H.Z. All authors have read and agreed to the published version of the manuscript.

**Funding:** This study was financially supported by the National Natural Science Foundation of China (No. 31971659).

**Data Availability Statement:** No new data were created or analyzed in this study. Data sharing is not applicable to this article.

**Conflicts of Interest:** The authors declare that the research was conducted in the absence of any commercial or financial relationships that could be construed as a potential conflict of interest.

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
