# Peer review of "Screen of Pinus massoniana for Resistance to Pinewood Nematode: In Vitro Propagation and Evaluation of Regenerated Microshoots"

_forests, doi:10.3390/f14051056_

Round 1

Reviewer 1 Report

This paper under review (Guo et al.) describes a method for creating micropropagation of Pinus massoniana and a method for evaluating pinewood nematode resistance using the culture tissue. The work is mostly well designed and well done. I do not find any problems in this study. I would like to suggest only a few minor changes before publication.

Specific comments are below.

-------------------------------------

Title:

This title is the same as the title of the paper No.44 in reference (only the tree species is difference). It feels like a transcription of the paper No.44, so I recommend changing it to a different title.

Title and Abstract:

pine wood nematode -> pinewood nematode

Results:

L303-305,

“The number of nematodes … significantly higher than that of all other clones (P<0.05) (Figure 6A).”

Figure 6A shows that the number of nematodes recovered from clone 253 was not statistically different from that recovered from clone 8.

Discussion:

L362-378,

The result of SSR analysis in this study is presented later in this paragraph (mutation occurred in 23.3% of shoots, with an average mutation rate per locus is 6.7%) and you state that the mutation risk in this method is low. What is the basis for stating that these values are low? You would have to show that the mutation rate in this study is low in comparison to other similar experiments.

I have one more question. It seemed that the mutation rate at locus P.Ma51 (16.7%) is remarkably higher than others. Can you see the similar trends in previous reports? If so, the readers will want to know why.

Reviewer 2 Report

The authors of this work were able to design and test a powerfull tool for Pinus massoniana propagation and performed important testing of its resistance against Bursaphelenchus xylophilus (PWN) nematode. I am finding the article well prepared and worth of publishing. Quality of the manuscript is high; I have only a couple of remarks:

line 130: please add isolate of what organism to make the text clearer for the reader

line 377: low mutation risk; I think this statement should be described in more detail

Conclusion: you should mention in this part aims of your next research (if there will be any), how will you use the clone 227 which showed tolerance to PWN, would it be possible eg. to use it in further breeding?
